

# Majorana lattice gauge theory: Symmetry breaking, topological order and intertwined orders all in one

Jian-Jian Miao⋆

Department of Physics, The Chinese University of Hong Kong,
Shatin, New Territories, Hong Kong, China.

⋆ jjmiao@phy.cuhk.edu.hk

## Abstract

The Majorana lattice gauge theory purely composed of Majorana fermions on square lattice is studied throughly. The ground state is obtained exactly and exhibits the coexistence of symmetry breaking and topological order. The $Z_2$ symmetry breaking of matter fields leads to the intertwined antiferromagnetic spin order and $\eta$-pairing order. The topological order is reflected in the $Z_2$ quantum spin liquid ground state of gauge fields. The Majorana lattice gauge theory, alternatively can be viewed as interacting Majorana fermion model, is possibly realized on a Majorana-zero-mode lattice.



## 1 Introduction

Landau's symmetry breaking theory establishes the first paradigm of phases of matters in condensed matter physics. Different phases are characterized by different symmetries. A phase

transition is determined how symmetry changes across the critical point and can be described by the local order parameters that transform nontrivially under the symmetry transformation. Landau's theory successfully accounts for the appearance of various low temperature orders due to spontaneous symmetry breaking, e.g. crystals and magnets. However, the discovery of fractional quantum Hall (FQH) effect [1] and high-$T_c$ superconductors [2] provides the phenomena beyond the paradigm of Landau's symmetry breaking theory. Anderson proposed the quantum spin liquid [3–9] (QSL) without symmetry breaking is the key to the mechanism of high-$T_c$ superconductivity [10]. Wen found different chiral spin liquids have exactly the same symmetry [11], as well as different FQH states. These new orders without symmetry breaking and local order parameters are characterized by the topological invariants, e.g. ground state degeneracy on the torus and nontrivial edge states [12]. Dubbed topological orders, these nontrivial gapped disordered phases possess long-range entanglement at zero temperature [13–15]. A new paradigm of topological phases of matters emerges and flourishes in the past decades [16]. A critical outstanding issue comes that is it possible to unify the symmetry breaking and topological order frameworks into a single formalism?

The Majorana fermion (MF) perspective of strongly correlated systems provide new insights into the issue. The MFs are real counterparts of complex fermions [17–19]. Not only can be realized experimentally, e.g. in the interface of s-wave superconductor and strong topological insulator [20–24], but also have the potential to implement topological quantum computation [25], MFs (Majorana zero modes) become the hot topics in condensed matter physics. Moreover the strongly interacting models [26–29] built from MFs can host exotic phenomena, such as Majorana dualities [30], the emergence of supersymmetry [31–33] and supersymmetry breaking [34], Majorana surface code [35,36], tricritical Ising point [37,38], topological order [39,40], SYK model with black hole physics [41,42].

In this paper, a novel Majorana lattice gauge theory is proposed, which can also be viewed as an interacting MF model. The ground state is obtained exactly with matter and gauge fields exhibiting symmetry breaking and topological order respectively. The $Z_2$ symmetry breaking leads to intertwined antiferromagnetic (AFM) spin and $\eta$-pairing orders characterized by local order parameters. The $Z_2$ topological order in the $Z_2$ QSL is characterized by the ground state degeneracy on torus. Even though matter and gauge fields are coupled in the ultraviolet limit, the decoupling of matter and gauge fields in the infrared limit leads to the nontrivial coexistence of symmetry breaking and topological order. The Majorana lattice gauge theory provides the first concrete example to unify symmetry breaking, topological order and intertwined orders.

## 2 Majorana lattice gauge theory

The building blocks of Majorana lattice gauge theory are MFs only. The MFs are described by real operators $\left(\gamma_{\mathbf{r}}^{j}\right)^{\dagger} = \gamma_{\mathbf{r}}^{j}$ obeying the anticommutation relations $\left\{\gamma_{\mathbf{r}}^{j}, \gamma_{\mathbf{r}'}^{j'}\right\} = 2\delta^{jj'}\delta_{\mathbf{rr}'}$ with site index $\mathbf{r}$ and flavor index $j = 1, \cdots, m$. Note the total MF flavors $m$ must be an even integer to ensure locality. For concreteness, a representative gauge theory on square lattice is considered. On each site the matter fields are four $\gamma$ MFs. The four $\gamma$ MFs can represent the conventional complex fermion operators $c_{\mathbf{r}s}$ describing the electrons with spin polarization $s = \uparrow, \downarrow$

$$c_{\mathbf{r}\uparrow} = \frac{1}{2}\left(\gamma_{\mathbf{r}}^{1} - i\gamma_{\mathbf{r}}^{2}\right), \tag{1a}$$

$$c_{\mathbf{r}\downarrow} = \frac{1}{2}\left(\gamma_{\mathbf{r}}^{3} - i\gamma_{\mathbf{r}}^{4}\right). \tag{1b}$$

The on-site interaction $H_U$ of $\gamma$ MFs is given by

$$H_U = \frac{U}{4} \sum_{\mathbf{r}} \left(i\gamma_{\mathbf{r}}^1 \gamma_{\mathbf{r}}^2\right)\left(i\gamma_{\mathbf{r}}^3 \gamma_{\mathbf{r}}^4\right) \tag{2a}$$

$$= U \sum_{\mathbf{r}} \left(n_{\mathbf{r}\uparrow} - \frac{1}{2}\right)\left(n_{\mathbf{r}\downarrow} - \frac{1}{2}\right), \tag{2b}$$

where $n_{\mathbf{r}s} = c_{\mathbf{r}s}^\dagger c_{\mathbf{r}s}$ is the electron density operator for spin $s$. The on-site interaction $H_U$ corresponds to the Hubbard interaction of electrons. Four $\chi$ MFs on each site are introduced and act as gauge fields. In the conventional lattice gauge theory, the gauge fields live on the links of lattice. As shown later, the product of two $\chi$ MFs on nearest neighbor sites correspond to the conventional bosonic gauge fields. Thus the total flavors of $\chi$ MFs are chosen to be equal to the coordination number $z = 4$ of the square lattice. Besides the on-site interaction $H_V$ with the same form as $\gamma$ MFs'

$$H_V = V \sum_{\mathbf{r}} \left(i\chi_{\mathbf{r}}^1 \chi_{\mathbf{r}}^2\right)\left(i\chi_{\mathbf{r}}^3 \chi_{\mathbf{r}}^4\right), \tag{3}$$

a plaquette interaction $H_K$ of $\chi$ MFs is also introduced

$$H_K = K \sum_{\mathbf{r}} \left(i\chi_{\mathbf{r}}^2 \chi_{\mathbf{r}}^1\right)\left(i\chi_{\mathbf{r}+\hat{x}}^3 \chi_{\mathbf{r}+\hat{x}}^2\right)\left(i\chi_{\mathbf{r}+\hat{x}+\hat{y}}^4 \chi_{\mathbf{r}+\hat{x}+\hat{y}}^3\right)\left(i\chi_{\mathbf{r}+\hat{y}}^1 \chi_{\mathbf{r}+\hat{y}}^4\right), \tag{4}$$

where each term includes four sites forming a plaquette on square lattice. Note all $\chi$ Majorana fermions within a plaquette are included in each plaquette interaction as shown in Figure 1, which to pin down the flavor indexes. The square lattice is divided into two dual sublattices $a$ and $b$, that is the plaquette centers of one sublattice correspond to the sites of another sublattice. The coupling between matter and gauge fields is given by

$$H_t = t \sum_{\langle \mathbf{r}_a \mathbf{r}_b \rangle} \left(i\gamma_{\mathbf{r}_a}^2 \gamma_{\mathbf{r}_b}^1 + i\gamma_{\mathbf{r}_a}^4 \gamma_{\mathbf{r}_b}^3\right)\left(i\chi_{\mathbf{r}_a}^j \chi_{\mathbf{r}_b}^k\right) \tag{5a}$$

$$= t \sum_{\langle \mathbf{r}_a \mathbf{r}_b \rangle s} \left(c_{\mathbf{r}_a s}^\dagger c_{\mathbf{r}_b s} + c_{\mathbf{r}_a s}^\dagger c_{\mathbf{r}_b s}^\dagger + H.c.\right)\left(i\chi_{\mathbf{r}_a}^j \chi_{\mathbf{r}_b}^k\right), \tag{5b}$$

where the quadratic $\gamma$ MFs correspond to nearest neighbor electron hopping and equal-spin-pairing, meanwhile the $\gamma$ MFs are coupled to $\chi^j$ and $\chi^k$ MFs connecting nearest neighbor sites as shown in Figure 1. The full Hamiltonian of the Majorana lattice gauge theory is the sum of above interactions

$$H = H_U + H_V + H_K + H_t, \tag{6}$$

which is also an interacting MF model. The Hamiltonian is constructed based on the locality principle, that is interactions are as local as possible. The full Hamiltonian has the global $Z_2$ symmetry by interchanging $\gamma$ MFs for all sites as follows

$$\gamma_{\mathbf{r}}^1 \leftrightarrow \gamma_{\mathbf{r}}^3, \tag{7a}$$

$$\gamma_{\mathbf{r}}^2 \leftrightarrow \gamma_{\mathbf{r}}^4, \tag{7b}$$

and also the local $Z_2$ gauge symmetry by interchanging $\gamma$ MFs on site $\mathbf{r}$ and $\chi$ MFs between site $\mathbf{r}$ and its four nearest neighbor sites are

$$\begin{cases} \gamma_{\mathbf{r}}^2 \to -\gamma_{\mathbf{r}}^2, & \gamma_{\mathbf{r}}^4 \to -\gamma_{\mathbf{r}}^4, \quad \mathbf{r} = \mathbf{r}_a, \\ \gamma_{\mathbf{r}}^1 \to -\gamma_{\mathbf{r}}^1, & \gamma_{\mathbf{r}}^3 \to -\gamma_{\mathbf{r}}^3, \quad \mathbf{r} = \mathbf{r}_b, \end{cases} \tag{8a}$$

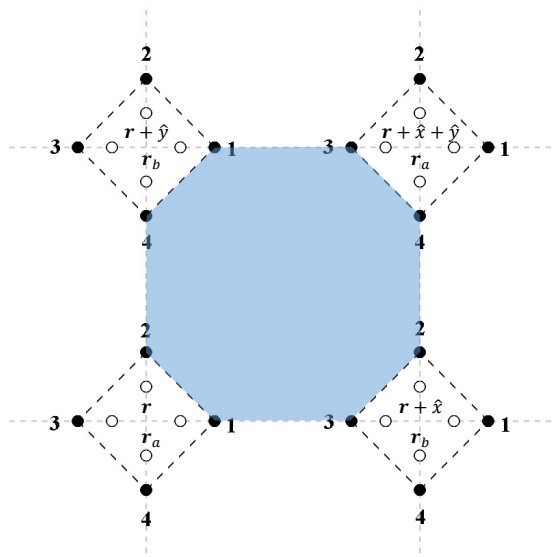

Figure 1: MFs on the square lattice. On each site, four $\gamma$ and four $\chi$ MFs are denoted as white and black points surrounding the site. The numbers $1-4$ denote the MF flavors. The square lattice is divided into two dual sublattices $a$ and $b$. The shadow region denotes the plaquette interaction $H_K$.

and

$$\chi_{\mathbf{r}}^1 \leftrightarrow \chi_{\mathbf{r}'}^3, \quad \mathbf{r}' = \mathbf{r} + \hat{x}, \tag{8b}$$

$$\chi_{\mathbf{r}}^3 \leftrightarrow \chi_{\mathbf{r}'}^1, \quad \mathbf{r}' = \mathbf{r} - \hat{x}, \tag{8c}$$

$$\chi_{\mathbf{r}}^2 \leftrightarrow \chi_{\mathbf{r}'}^4, \quad \mathbf{r}' = \mathbf{r} + \hat{y}, \tag{8d}$$

$$\chi_{\mathbf{r}}^4 \leftrightarrow \chi_{\mathbf{r}'}^2, \quad \mathbf{r}' = \mathbf{r} - \hat{y}. \tag{8e}$$

Note the novel Majorana lattice gauge theory is composed of MFs only, which is different from the conventional gauge theory composed of MFs on sites and spins on links such as [43].

## 3 $V = 0$: exactly solvable model and symmetry breaking of matter fields

In the limit $V = 0$, the Majorana lattice gauge theory on square lattice reduces to an exactly solvable model $H_0 = H_U + H_K + H_t$. The on-site interaction $H_U$ is obviously composed of commuting projectors, meanwhile the plaquette interaction $H_K$ is also composed of commuting projectors as two neighbor plaquettes share two $\chi$ MFs. Note in the coupling $H_t$, $\gamma^1$ and $\gamma^3$ on sublattice $a$ meanwhile $\gamma^2$ and $\gamma^4$ on sublattice $b$ are absent, we define the $\gamma$ MF *site operators*

$$\hat{C}_{\mathbf{r}} = \begin{cases} i\gamma_{\mathbf{r}_a}^1 \gamma_{\mathbf{r}_a}^3, & \mathbf{r} = \mathbf{r}_a, \\ i\gamma_{\mathbf{r}_b}^2 \gamma_{\mathbf{r}_b}^4, & \mathbf{r} = \mathbf{r}_b. \end{cases} \tag{9}$$

Since $\left[\hat{C}_{\mathbf{r}}, H_0\right] = 0$, the site operators are constants of motion in the limit $V = 0$. Also $\hat{C}_{\mathbf{r}}^2 = 1$, the eigenvalues of the site operators take $Z_2$ values $C_{\mathbf{r}} = \pm 1$. The on-site interaction $H_U$ can

be written in terms of site operators as

$$H_U = -\frac{U}{4}\left[\sum_{\mathbf{r}_a} \hat{C}_{\mathbf{r}_a}\left(i\gamma^2_{\mathbf{r}_a}\gamma^4_{\mathbf{r}_a}\right) + \sum_{\mathbf{r}_b} \hat{C}_{\mathbf{r}_b}\left(i\gamma^1_{\mathbf{r}_b}\gamma^3_{\mathbf{r}_b}\right)\right].\tag{10}$$

Similarly we define the $\chi$ MF *bond operators*

$$\hat{D}_{\mathbf{r},\mathbf{r}'} = -\hat{D}_{\mathbf{r}',\mathbf{r}} = \begin{cases} i\chi^1_{\mathbf{r}}\chi^3_{\mathbf{r}'}, & \mathbf{r}' = \mathbf{r}+\hat{x}, \\ i\chi^2_{\mathbf{r}}\chi^4_{\mathbf{r}'}, & \mathbf{r}' = \mathbf{r}+\hat{y}. \end{cases}\tag{11}$$

As $\left[\hat{D}_{\mathbf{r},\mathbf{r}'}, H_0\right] = 0$ and $\hat{D}^2_{\mathbf{r},\mathbf{r}'} = 1$, the bond operators are also constants of motion in the limit $V = 0$ with $Z_2$ eigenvalues $D_{\mathbf{r},\mathbf{r}'} = \pm 1$. The plaquette interaction $H_K$ and coupling $H_t$ can be written in terms of bond operators as

$$H_K = -K\sum_{\mathbf{r}} \hat{D}_{\mathbf{r},\mathbf{r}+\hat{x}}\hat{D}_{\mathbf{r}+\hat{x},\mathbf{r}+\hat{x}+\hat{y}}\hat{D}_{\mathbf{r}+\hat{x}+\hat{y},\mathbf{r}+\hat{y}}\hat{D}_{\mathbf{r}+\hat{y},\mathbf{r}},\tag{12}$$

$$H_t = t\sum_{\langle \mathbf{r}_a\mathbf{r}_b\rangle} \left(i\gamma^2_{\mathbf{r}_a}\gamma^1_{\mathbf{r}_b} + i\gamma^4_{\mathbf{r}_a}\gamma^3_{\mathbf{r}_b}\right)\hat{D}_{\mathbf{r}_a,\mathbf{r}_b}.\tag{13}$$

Therefore in the limit $V = 0$ the Majorana lattice gauge theory reduces to the quadratic $\gamma$ MFs coupled to the static $Z_2$ gauge fields. The constants of motion $\hat{C}_{\mathbf{r}}$ and $\hat{D}_{\mathbf{r},\mathbf{r}'}$ serve as static $Z_2$ gauge fields. The exact solvability of the model $H_0$ is in the same spirit of the exactly solvable Kitaev honeycomb model [44].

We can replace the operators $\hat{C}_{\mathbf{r}}$ and $\hat{D}_{\mathbf{r},\mathbf{r}'}$ by their eigenvalues in $H_0$ and the ground states of $H_0$ is determined by the configurations $\{C_{\mathbf{r}}\}$ and $\{D_{\mathbf{r},\mathbf{r}'}\}$ with lowest energy. The plaquette interaction $H_K$ is of the same form of Wegner's Ising lattice gauge theory [45], where the $Z_2$ eigenvalues $D_{\mathbf{r},\mathbf{r}'} = \pm 1$ act as classical Ising spins. We define the *local gauge transformation $G_{\mathbf{r}}$* on site $\mathbf{r}$ that only the $D_{\mathbf{r},\mathbf{r}'}$ connecting to site $\mathbf{r}$ change sign

$$G_{\mathbf{r}} : D_{\mathbf{r},\mathbf{r}'} \to -D_{\mathbf{r},\mathbf{r}'},\tag{14a}$$

meanwhile the $\gamma$ MFs on site $\mathbf{r}$ change as

$$\begin{cases} \gamma^2_{\mathbf{r}_a} \to -\gamma^2_{\mathbf{r}_a}, & \gamma^4_{\mathbf{r}_a} \to -\gamma^4_{\mathbf{r}_a}, & \mathbf{r} = \mathbf{r}_a, \\ \gamma^1_{\mathbf{r}_b} \to -\gamma^1_{\mathbf{r}_b}, & \gamma^3_{\mathbf{r}_b} \to -\gamma^3_{\mathbf{r}_b}, & \mathbf{r} = \mathbf{r}_b, \end{cases}\tag{14b}$$

which is just the local $Z_2$ gauge symmetry in Eq. 8. The Hamiltonian $H_0$ is gauge invariant, thus the ground state configurations $\{D_{\mathbf{r},\mathbf{r}'}\}$ include $2^N$ configurations that can be related to the uniform configuration $\{D_{\mathbf{r},\mathbf{r}'} = 1\}$[1] by all local gauge transformations, where $N$ is the total number of sites, i.e. the total number of local gauge transformations. Note the local gauge transformations won't alter the $\gamma$ MF site operators. To determine the ground state configurations $\{C_{\mathbf{r}}\}$, the large-$U$ limit $|U| \gg |t|$ is firstly considered to gain intuitive physical understanding. The large-$U$ limit enforces the two low-energy states of $\gamma$ MFs on each site as $i\gamma^2_{\mathbf{r}_a}\gamma^4_{\mathbf{r}_a} = \text{sign}U C_{\mathbf{r}_a} = \pm 1$ if site belongs to $a$ sublattice or $i\gamma^1_{\mathbf{r}_b}\gamma^3_{\mathbf{r}_b} = \text{sign}U C_{\mathbf{r}_b} = \pm 1$ if site belongs to $b$ sublattice, where $\text{sign}U$ is the sign of on-site interaction strength $U$. Thus the ground state configurations $\{C_{\mathbf{r}}\}$ must be within the low-energy subspace of $2^N$ direct product states. In the limit $|U| \gg |t|$, the coupling $H_t$ can be treated as perturbation and the perturbation theory is adopted to derive the effective Hamiltonian within the low-energy subspace

$$H^{\text{eff}} = -\frac{t^2}{|U|}\sum_{\langle \mathbf{r}_a\mathbf{r}_b\rangle} \left(i\gamma^2_{\mathbf{r}_a}\gamma^1_{\mathbf{r}_b}\right)\left(i\gamma^4_{\mathbf{r}_a}\gamma^3_{\mathbf{r}_b}\right)\hat{D}^2_{\mathbf{r}_a,\mathbf{r}_b} = \frac{t^2}{|U|}\sum_{\langle \mathbf{r}_a\mathbf{r}_b\rangle} C_{\mathbf{r}_a}C_{\mathbf{r}_b},\tag{15}$$

---

[1] For positive $K$, the representative ground state configuration is uniform configuration. For negative $K$, the representative ground state configuration is $\{D_{\mathbf{r}_a,\mathbf{r}_a+\hat{x}} = -1, D_{\mathbf{r}_b,\mathbf{r}_b+\hat{x}} = D_{\mathbf{r}+\hat{y}} = 1\}$.

where only even orders survive and the lowest nontrivial terms come from second order. The effective Hamiltonian is the AFM Ising model. Note even the $\gamma$ MFs and $\chi$ MF bond operators are coupled in the ultraviolet limit, the effective Hamiltonian is independent of bond operators. Thus the $\gamma$ and $\chi$ MFs are deoupled in the infrared limit due to the $Z_2$ characteristic of bond operators. The AFM Ising model indicates the ground state configurations are $\{C_{\mathbf{r}_a} = -C_{\mathbf{r}_b} = \pm 1\}$ with two-fold degeneracy in the positive large-$U$ limit and high order terms won't lift the degeneracy. For generic interaction strength $U$, the ground state configurations $\{C_{\mathbf{r}}\}$ can be determined numerically by diagonalizing the quadratic $\gamma$ MFs on finite size lattice and searching for the configurations with lowest ground state energy. The exact two-fold degeneracy of ground state configurations $\{C_{\mathbf{r}_a} = -C_{\mathbf{r}_b} = \pm 1\}$ is numerically confirmed for arbitrary $U$ in the Appdendix A. Once the ground state configurations are pinned down, the orders in the ground states can be identified.

The two-fold degeneracy in terms of $\gamma$ MF site operators indicates the $Z_2$ symmetry breaking of matter fields in the ground states of of $H_0$. Recall the global $Z_2$ symmetry of the full Hamiltonian $H$ in Eq. 7, under which the site operators change sign $\hat{C}_{\mathbf{r}} \leftrightarrow -\hat{C}_{\mathbf{r}}$. Thus the expectation values of $\hat{C}_{\mathbf{r}}$ naturally serve as local order parameters. Nevertheless we shall define more physical local order parameters in terms of electron operators by introducing the spin and charge operators

$$\hat{S}_{\mathbf{r}}^{\alpha} = \frac{1}{2} \begin{pmatrix} c_{\mathbf{r}\uparrow}^{\dagger} & c_{\mathbf{r}\downarrow}^{\dagger} \end{pmatrix} \tau^{\alpha} \begin{pmatrix} c_{\mathbf{r}\uparrow} \\ c_{\mathbf{r}\downarrow} \end{pmatrix}, \tag{16}$$

$$\hat{Q}_{\mathbf{r}}^{\alpha} = \frac{1}{2} \begin{pmatrix} c_{\mathbf{r}\uparrow}^{\dagger} & c_{\mathbf{r}\downarrow} \end{pmatrix} \tau^{\alpha} \begin{pmatrix} c_{\mathbf{r}\uparrow} \\ c_{\mathbf{r}\downarrow}^{\dagger} \end{pmatrix}, \tag{17}$$

where $\tau^{\alpha}$ are Pauli matrices with $\alpha = x, y, z$. Note the operator $2\hat{Q}_{\mathbf{r}}^{z} = n_{\mathbf{r}\uparrow} + n_{\mathbf{r}\downarrow} - 1$ measures the charges with respect to half-filling. Since

$$\hat{S}_{\mathbf{r}_a}^{y} + \hat{Q}_{\mathbf{r}_a}^{y} = -\frac{1}{2}\hat{C}_{\mathbf{r}_a}, \tag{18}$$

$$\hat{S}_{\mathbf{r}_b}^{y} - \hat{Q}_{\mathbf{r}_b}^{y} = -\frac{1}{2}\hat{C}_{\mathbf{r}_b}, \tag{19}$$

the $y$-components of spin and charge operators also transform nontrivially under $Z_2$ symmetry transformation and the their expectation values $S_{\mathbf{r}}^{y} = \langle \hat{S}_{\mathbf{r}}^{y} \rangle$ and $Q_{\mathbf{r}}^{y} = \langle \hat{Q}_{\mathbf{r}}^{y} \rangle$ serve as local order parameters. Without loss of generality, the local order parameters are calculated under the uniform configuration $\{D_{\mathbf{r},\mathbf{r}'} = 1\}$ which differs other configurations by local gauge transformation. In the uniform configuration the model $H_0$ is equivalent to the BCS-Hubbard model at the exactly solvable point [46], and the local order parameters are given by

$$S_{\mathbf{r}}^{y} = \pm (-)^{\mathbf{r}} \frac{1}{4} \left( 1 + \frac{1}{N} \sum_{\mathbf{k}}' \frac{U}{E_k} \right), \tag{20}$$

$$Q_{\mathbf{r}}^{y} = \pm \frac{1}{4} \left( 1 - \frac{1}{N} \sum_{\mathbf{k}}' \frac{U}{E_k} \right), \tag{21}$$

where $E_k = \frac{1}{2}\sqrt{U^2 + 16t^2 \epsilon_{\mathbf{k}}^2}$ is the quasiparticle dispersion of $\gamma$ MFs and $\epsilon_{\mathbf{k}} = 2(\cos k_x + \cos k_y)$ is the form factor on square lattice.[2] The summation $\sum_{\mathbf{k}}'$ is over the magnetic Brillouin zone, which is half of square lattice Brillouin zone. $S_{\mathbf{r}}^{y}$ and $Q_{\mathbf{r}}^{y}$ characterize the spin and pairing orders respectively. The staggered factor in $S_{\mathbf{r}}^{y}$ indicates the spin order is the AFM order. Recall

---

[2]For negative $K$, the quasiparticle dispersion and form factor change to $E_k^{-} = \frac{1}{2}\sqrt{U^2 + 16t^2 |\gamma_{\mathbf{k}}^{-}|^2}$ and $\gamma_{\mathbf{k}}^{-} = 2(-i\sin k_x + \cos k_y)$ respectively.

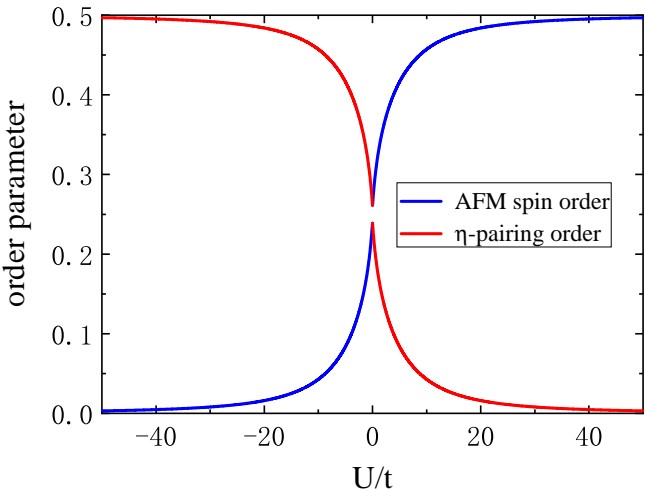

Figure 2: The magnitudes of local order parameters $\left|S_{\mathbf{r}}^{y}\right|$ and $\left|Q_{\mathbf{r}}^{y}\right|$ as function of $U/t$. The singular point $U = 0$ indicates the gap close of $E_k$. In the large positive/negative $U$ limit, the magnitude of spin/pairing order saturates.

the definition $Q_{\mathbf{r}}^{y} = \frac{i}{2}\left\langle c_{\mathbf{r}\downarrow}c_{\mathbf{r}\uparrow} - c_{\mathbf{r}\uparrow}^{\dagger}c_{\mathbf{r}\downarrow}^{\dagger}\right\rangle$, the pairing order is the spin-singlet $\eta$-pairing [47,48]. The two orders coexist as they break the same $Z_2$ symmetry. However the repulsive Hubbard interaction favors the spin order while the attractive Hubbard interaction favors the pairing order. Thus the two orders also compete with each other that lead to the waxing and waning pattern of magnitudes of local order parameters in Figure 2. Such coexistence and competition of orders in an exactly solvable model provide a concrete example of intertwined orders in strongly correlated electron systems [49].

## 4 $V \neq 0$: exact ground state and topological order of gauge fields

The on-site interaction $H_V$ spoils the exact solvability of $H_0$ as $\left\{\left(i\chi_{\mathbf{r}}^{1}\chi_{\mathbf{r}}^{2}\right)\left(i\chi_{\mathbf{r}}^{3}\chi_{\mathbf{r}}^{4}\right), \hat{D}_{\mathbf{r},\mathbf{r}'}\right\} = 0$. To gain intuitive understanding of physical effect of $H_V$, the large-$V$ limit is firstly considered. As $\left(\chi_{\mathbf{r}}^{1}\chi_{\mathbf{r}}^{2}\chi_{\mathbf{r}}^{3}\chi_{\mathbf{r}}^{4}\right)^{2} = 1$, in the limit $|V| \gg |K|$ the two low-energy states of $\chi$ MFs on each site are identified as $\chi_{\mathbf{r}}^{1}\chi_{\mathbf{r}}^{2}\chi_{\mathbf{r}}^{3}\chi_{\mathbf{r}}^{4} = \mathrm{sign}V$, where $\mathrm{sign}V$ is the sign of on-site interaction strength $V$. Thus the on-site interaction $H_V$ can be viewed as local constraints in the large-$V$ limit. We define the Pauli spin operators in terms of $\chi$ MFs

$$\sigma_{\mathbf{r}}^{x} = i\chi_{\mathbf{r}}^{1}\chi_{\mathbf{r}}^{2} = \mathrm{sign}V i\chi_{\mathbf{r}}^{4}\chi_{\mathbf{r}}^{3}, \tag{22a}$$

$$\sigma_{\mathbf{r}}^{y} = i\chi_{\mathbf{r}}^{1}\chi_{\mathbf{r}}^{3} = \mathrm{sign}V i\chi_{\mathbf{r}}^{2}\chi_{\mathbf{r}}^{4}, \tag{22b}$$

$$\sigma_{\mathbf{r}}^{z} = \mathrm{sign}V i\chi_{\mathbf{r}}^{1}\chi_{\mathbf{r}}^{4} = i\chi_{\mathbf{r}}^{3}\chi_{\mathbf{r}}^{2}, \tag{22c}$$

where the local constraints $\chi_{\mathbf{r}}^{1}\chi_{\mathbf{r}}^{2}\chi_{\mathbf{r}}^{3}\chi_{\mathbf{r}}^{4} = \mathrm{sign}V$ are used in the second equality. The four $\chi$ MFs with local constraints give a faithful representation of Pauli spin operators, such as the identity $\sigma_{\mathbf{r}}^{x}\sigma_{\mathbf{r}}^{y}\sigma_{\mathbf{r}}^{z} = \mathrm{sign}V i\chi_{\mathbf{r}}^{1}\chi_{\mathbf{r}}^{2}\chi_{\mathbf{r}}^{3}\chi_{\mathbf{r}}^{4} = i$, and the two low-energy states on each site also match the two-dimensional Hilbert space of Pauli spin. The plaquette interaction $H_K$ in the Pauli spin representation is given by

$$H_K = -K\left(\mathrm{sign}V\right)^{2}\sum_{\mathbf{r}}\sigma_{\mathbf{r}}^{x}\sigma_{\mathbf{r}+\hat{x}}^{z}\sigma_{\mathbf{r}+\hat{x}+\hat{y}}^{x}\sigma_{\mathbf{r}+\hat{y}}^{z}, \tag{23}$$

which is the exactly solvable Wen plaquette model [50], and is equivalent to the toric code [51] hosting the exact $Z_2$ QSL ground state with topological order. The ground states of Wen plaquette model on torus is topologically four-fold degeneracy. Note different sign$V$'s lead to the same topological order.

In the spirit of Noether's theorem, the local gauge symmetry has a corresponding conserved gauge charge and vice versa. Thus the $Z_2$ gauge symmetry in Eq. 14 leads to the conservation of $Z_2$ gauge charge $\hat{P}_{\mathbf{r}} = \hat{M}_{\mathbf{r}} \hat{G}_{\mathbf{r}}$, where $\hat{M}_{\mathbf{r}} = \gamma_{\mathbf{r}}^1 \gamma_{\mathbf{r}}^2 \gamma_{\mathbf{r}}^3 \gamma_{\mathbf{r}}^4$ and $\hat{G}_{\mathbf{r}} = \chi_{\mathbf{r}}^1 \chi_{\mathbf{r}}^2 \chi_{\mathbf{r}}^3 \chi_{\mathbf{r}}^4$ characterize the fermion number parities of $\gamma$ and $\chi$ MFs on each site respectively. The properties $\left[H, \hat{P}_{\mathbf{r}}\right] = 0$ and $\hat{P}_{\mathbf{r}}^2 = 1$ manifest the $Z_2$ characteristic of conserved gauge charge $P_{\mathbf{r}} = \pm 1$. Even though separate fermion number parities of $\gamma$ and $\chi$ MFs are not conserved due to the coupling $H_t$, their total fermion number parity $P_{\mathbf{r}}$ is conserved in the ultraviolet limit. In $V \neq 0$ the site operators are still constants of motion but the bond operators are not. In the eigenbasis of bond operators, the signs $D_{\mathbf{r},\mathbf{r}'} = \pm 1$ in the coupling $H_t$ in Eq. 13 can be absorbed by the local gauge transformation. Thus the physics of $\gamma$ MFs, that is symmetry breaking and intertwined orders of matter fields, is unchanged for $V \neq 0$. Moreover, as shown in the limit $|U| \gg |t|$ in Eq. 15 the $\gamma$ and $\chi$ MFs are explicitly decoupled in the infrared limit due to $\hat{D}_{\mathbf{r},\mathbf{r}'}^2 = 1$. For more generic parameters $U$ and $t$, on the one hand, as the matter fields have discrete symmetry breaking, the low-energy excitations of matter fields are all gapped thus do not influence the gauge fields in the infrared limit, on the other hand, the matter fields only feel the gauge invariant quantities of gauge fields, i.e. the flux of plaquette, the plaquette interaction $H_K$ fixes the $Z_2$ flux of gauge fields and the only low-energy excitations of gauge fields are gapped visons, thus do not influence the matter fields in the infrared limit. The decoupling of between matter and gauge degrees of freedom in the infrared limit is essentially unique to Majorana/$Z_2$ lattice gauge theory. The low-energy effective Hamiltonian of gauge fields can be captured by $H_K + H_V$.[3] In the small-$V$ limit $|V| \ll |K|$, the ground state of gauge fields must lie in the $2^N$ ground state configurations of plaquette interaction $H_K$. The $2^N$ configurations are related to the uniform configuration $\{D_{\mathbf{r},\mathbf{r}'} = 1\}$ by all local gauge transformations. Note the operators $\hat{G}_{\mathbf{r}} = \chi_{\mathbf{r}}^1 \chi_{\mathbf{r}}^2 \chi_{\mathbf{r}}^3 \chi_{\mathbf{r}}^4$ constituting the on-site interaction $H_V$ implements the local gauge transformation $G_{\mathbf{r}}$ in Eq. 14. Thus the ground state of gauge fields is the equal weight linear superposition of $2^N$ gauge equivalent configurations, which is the famous Anderson's resonating valence bond state of QSL [3, 10]. In the ground state $\hat{G}_{\mathbf{r}} = \chi_{\mathbf{r}}^1 \chi_{\mathbf{r}}^2 \chi_{\mathbf{r}}^3 \chi_{\mathbf{r}}^4 = 1$, which is identical to the local constraint in the limit $|V| \gg |K|$ wherein the sign$V$ is irrelevant to ground state. The QSL ground state in the small-$V$ limit is adiabatically connected to the $Z_2$ QSL ground state in the large-$V$ limit. Even though the exact solvability of $H_0$ is spoiled by on-site interaction $H_V$, the exact ground state of $H$ is still extracted. The conservation of fermion number parity $\hat{G}_{\mathbf{r}} = 1$ of $\chi$ MFs in the exact ground state is another manifestation of decoupling of matter and gauge fields in the infrared limit. The conserved $Z_2$ gauge charge $\hat{G}_{\mathbf{r}} = 1$ corresponds to the local gauge symmetry of gauge fields only.

The zero-temperature phase diagram of Majorana lattice gauge theory for $V \neq 0$ is sketched in Figure 3. Besides the irrelevance of sign$V$, the sign of coupling constant $t$ is also irrelevant to ground state due to the bipartiteness of square lattice and we set $t > 0$. In regards of matter fields, the AFM spin order dominates for positive $U$, while the $\eta$-pairing order dominates for negative $U$. The singular point $U = 0$ indicates the macroscopic degeneracy due to all local constants of motion, i.e. all site operators, but not a critical point. The separation line $U/t = 0$ in the phase diagram is not a phase boundary but only a crossover. As for gauge fields, the ground state is $Z2A$ and $Z2B$ QSL for positive and negative $K$ respectively, which

---

[3]In principle, the matter fields can be integrated to derive the effective Hamiltonian of gauge fields. The coupling $H_t$ will generate effective interaction of bond operators. The local gauge symmetry $G_{\mathbf{r}}$ requires the bond operators in effective interaction form closed loop. The most dominant effective interaction is just the plaquette interaction $H_K$.

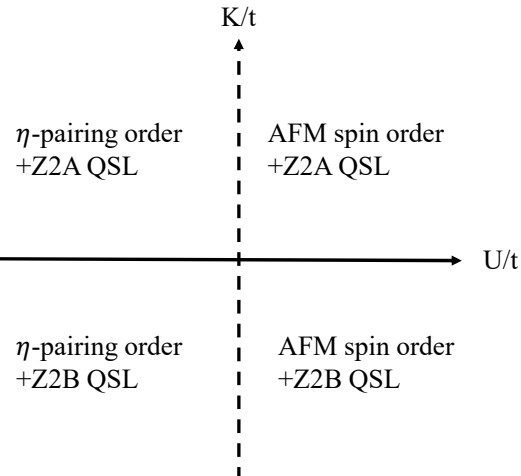

Figure 3: Zero-temperature phase diagram of Majorana lattice gauge theory for $V \neq 0$. The dashed line $U/t = 0$ denotes a crossover. The solid line $K/t = 0$ denotes a topological phase transition.

terminology is from the projective symmetry group classification [50]. The essential difference between $Z2A$ and $Z2B$ QSL is the $Z_2$ flux $F_{\mathbf{r}} = \sigma_{\mathbf{r}}^x \sigma_{\mathbf{r}+\hat{x}}^z \sigma_{\mathbf{r}+\hat{x}+\hat{y}}^x \sigma_{\mathbf{r}+\hat{y}}^z = \pm 1$ constituting the plaquette interaction $H_K$ in Eq. 23. The phase transition line $K/t = 0$ separating $Z2A$ and $Z2B$ phases is of first order phase transition, akin to the magnetic field induced first order phase transition of Ising model across the zero magnetic field line. However, different from that the magnetic field breaks the $Z_2$ symmetry of Ising model, the plaquette interaction keeps the local $Z_2$ gauge symmetry of gauge fields. The first order phase transition is topological phase transition, which means the phase transition has nothing to do with symmetry but only the discrete gauge structure of topological order, i.e. $Z_2$ flux, changes across the phase transition line. The physics of symmetry breaking and intertwined orders of matter fields is unchanged across the phase transition line, and the only subtle changes are the quasiparticle dispersion and form factor due to the sign change of $K$. Such first order topological phase transition is unchanged in the presence of matter fields as matter and gauge fields still decouple in the infrared limit across the phase transition line.

## 5   Discussion

The Majorana lattice gauge theory with $\gamma$ and $\chi$ MFs act as matter and gauge fields respectively on square lattice is studied throughly. The matter fields with symmetry breaking exhibit intertwined AFM spin order and $\eta$-pairing order, both of which break the same $Z_2$ symmetry meanwhile compete with each other. The gauge fields form $Z_2$ QSL ground state with $Z_2$ topological order therein. The unexpected coexistence of symmetry breaking and topological order is due to the decoupling of matter and gauge fields in the infrared limit, which is unique to Majorana/$Z_2$ lattice gauge theory and can't be straightforwardly generalized to other lattice gauge theories with discrete $Z_N$ ($N > 2$) or continuous symmetries. Formally, the global symmetry of matter fields can be spontaneously broken that leads to local order parameters. However the local gauge symmetry of gauge fields can never be broken according to Elitzur theorem [52,53] but can host topological order. The Majorana lattice gauge theory can unify these two frameworks in a single formalism.

The Majorana lattice gauge theory on square lattice is equivalent to Wegner's Ising gauge theory, AFM Ising model, BCS-Hubbard model, Wen plaquette model and toric code in various limits. On the other hand, Majorana lattice gauge theory can be easily generalized to other lattices and higher dimensions. Simply let the number of $\chi$ MFs on each site equals to the site coordination number $z$. We can also introduce $2m$ $\gamma$ MFs with $m > 2$ to include more degrees of freedom besides charge and spin, e.g. orbit. The global discrete $Z_2$ symmetry can also be promoted to global continuous $U(1)$ symmetry with particle number conservation and such systems can host phases such as deconfined phase with gapless Dirac fermions, orthogonal metal, and so on [54–57]. In principle the Majorana lattice gauge theory can harbor more exotic coexistence of different symmetry breaking and topological order.

The Majorana lattice gauge theory purely composed of MFs can be alternatively viewed as an interacting MF model. Recently, the Majorana-zero-mode lattice has been realized in a tunable way [58], which provides a natural platform to implement interacting MF model composed of local interactions only. Experimentally, the $Z_2$ domain wall as symmetry defect of $Z_2$ symmetry breaking can be detected by local probe, e.g. STM. Theoretically, the topological order is reflected in the ground state degeneracy on nontrivial base manifold [59]. A more realistic scheme is the detection of vison excitation of $Z_2$ topological order [60]. Future direction of research is the doping effect, that is doping on $\gamma$ or $\chi$ MFs to study the effect on AFM spin order or QSL, which may provide clues to high-$T_c$ cuprates.

Utilizing MF representation of matter and gauge fields instead of conventional complex fermions and bosonic gauge fields is crucial from the perspective of locality, indistinguishability and symmetry. First, the current interacting MF model on square lattice is not a simple coupling between BCS-Hubbard model and toric code. Only in the infinite $V$ limit, the interaction of $\chi$ MFs is exactly identical to the toric code. However, the coupling between $\gamma$ and $\chi$ MFs will then become a nonlocal interaction between electrons of BCS-Hubbard model and spins of toric code, which violates the locality principle. Even though using the unfamiliar MF representation, various limits are examined to make connection with related works. Second, the realization of the model in conception is using the Majorana zero modes in vortex lattice. Thus it is natural and necessary to formulate the theory in terms of MFs. The MF representation of matter fields, which puts spin and charge degrees of freedom on equal footing, provides a clear physical picture of intertwined orders. Writing gauge fields in terms of MFs provides a novel fractionalization routine of conventional bosonic gauge fields. Also the MF representation put matter and gauge fields on equal footing. The MF duality is explored in [40, 61] according to the indistinguishability of MFs, it is possible to seek duality between matter and gauge fields in the MF representation as in principle MFs are also identical particles like electrons. Third, for systems with $2n$ MF flavors in total, the maximal symmetry of the system can be directly read out as $SO(2n)$, such as Hubbard model at half-filling with $SO(4)$ symmetry. Thus it is more straightforward to perform the symmetry analysis in terms of MFs.

# A  Numerical confirmation of the exact two-fold degeneracy

For generic interaction strength $U$, the ground state configurations $\{C_{\mathbf{r}}\}$ are determined numerically. There are $2^N$ configurations $\{C_{\mathbf{r}}\}$ and $2^N$ local gauge transformation related configurations $\{D_{\mathbf{r},\mathbf{r}'}\}$ in total to complete the numerical traversal, where $N$ is the number of total sites. In consideration of numerical resources and time, first the $2^N$ configurations $\{C_{\mathbf{r}}\}$ under the uniform configuration $\{D_{\mathbf{r},\mathbf{r}'} = 1\}$ is explored up to lattice size $N = 4 \times 4$. The ground state configurations are indeed $\{C_{\mathbf{r}_a} = -C_{\mathbf{r}_b} = \pm 1\}$ with the exact two-fold degeneracy. Similar calculation is performed in Ref. [61]. Then an arbitrary local gauge transformation (corresponding to a numerically generated random integer number between 1 and $2^N$) is implemented

to generate another non-uniform configuration $\{D_{\mathbf{r},\mathbf{r}'}\}$ followed by the same exploration of $2^N$ configurations $\{C_{\mathbf{r}}\}$. The ground state configurations are still $\{C_{\mathbf{r}_a} = -C_{\mathbf{r}_b} = \pm 1\}$ with the exact two-fold degeneracy.

# Acknowledgement

JJM is grateful for the suggestions and comments from Yi Zhou. This work is supported by General Research Fund Grant No. 14302021 from Research Grants Council and Direct Grant No. 4053416 from the Chinese University of Hong Kong.

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
