# Peer review of "Majorana lattice gauge theory: symmetry breaking, topological order and intertwined orders all in one"

_SciPost Physics, doi:SciPost Phys. 14, 111 (2023)_

## Round 1 · Referee Report · Anonymous (Referee 1) · 2022-12-22

Strengths

1 - The model introduced is new and seems to support a variety of non-trivial physics.
2 - An analysis is carried out in various limits, making connections to previously-studied models .

Weaknesses

1 - The direction of the paper is a little difficult to follow. Suggestions for how to improve this are discussed below.

Report

This paper presents a model of Majorana fermions, for which the author claims hosts a variety of non-trivial phases. As the author notes, this construction is rather general, with clear generalizations to other geometries. Some interesting results are obtained in exactly solvable limits.

On page 7, the author states "For generic interaction strength U, the ground state configurations {C_r} can be determined numerically by diagonalizing the quadratic MFs on finite size lattice and searching for the configurations with lowest ground state energy." It would be helpful if the author included more details on this - specifically, what configuration of bond operators D were used in this calculation? Is there a range of parameters where this exact two-fold degeneracy does not hold?

While written in terms of Majorana fermions, this model can also be written in terms of complex fermions coupled to a Z2 gauge field. Unless I misunderstand, I believe that this would give the BCS-Hubbard model studied in [47], coupled to a toric code. Regardless, I believe that expressing the model in this language at least once would be helpful to readers who are more familiar with such systems and also contextualize the model with related works. This also begs the question of why writing both the matter and gauge fields purely in terms of Majorana fermions is important. Solvability of the BCS-Hubbard model requires a Majorana description of the matter fields, but what utility does a Majorana description of the gauge fields give the reader? The motivation for this representation should be made more clear in the paper - as keeping track of the four different flavors of Majorana fermions can be difficult to follow, it is important to have a clear motivation for why this notation is useful.

The author briefly mentions a topological phase transition at K/t=0, corresponding to the change between the Z2A and Z2B nature of the gauge fields, but does not elaborate on it further. I would suggest that more be said about this transition - is it clear what this transition is in the limit that the matter and gauge fields decouple? If so, is it unchanged by the presence of matter fields?

The author mentions the decoupling between matter and gauge degrees of freedom in the IR several times, but only shows this phenomenon explicitly in the limit U >> t. As I understand the paper, this feature is one of the central results, and a more explicit discussion of whether it hols for more general U/t would be helpful.

In the large-U analysis on page 7, only positive U is considered. It would seem that a large and negative U limit is also accessible perturbatively - does this analysis agree with later results?

In summary, the author presents a new and interesting model of interacting fermions which is believed to host a variety of non-trivial physics. A study of this model is a noteworthy contribution to the field and is suitable for publication. However, I found the presentation a little difficult to follow, and include some suggestions in "Requested changes" on how to improve readability.

Requested changes

1 - Typo at the end of page 13, "puryly" should be spelled "purely."
2 - On page 9, define more clearly what is meant by a "folded Brillouin zone."
3 - Include more details on the numerical diagonalization discussed in page 7.
4 - A clear motivation for why this Majorana representation of the gauge fields is important should be stated.
I would suggest the following changes in order to make the paper easier to follow; however, the author is free to disagree with this.
5 - The discussion of symmetries of the model, both local and global, should be presented at the very beginning of the paper when the Hamiltonian is introduced.

  • validity: good
  • significance: good
  • originality: high
  • clarity: ok
  • formatting: good
  • grammar: reasonable

Author:  Jian-Jian Miao  on 2023-01-14  [id 3235]

(in reply to Report 1 on 2022-12-22)

This paper presents a model of Majorana fermions, for which the author claims hosts a variety of non-trivial phases. As the author notes, this construction is rather general, with clear generalizations to other geometries. Some interesting results are obtained in exactly solvable limits.

Re: I’m appreciative of the referee’s constructive comments and suggestions. I have responded all the constructive suggestions made by the referee, enclosed please find my point-to-point response.

On page 7, the author states "For generic interaction strength U, the ground state configurations {C_r} can be determined numerically by diagonalizing the quadratic MFs on finite size lattice and searching for the configurations with lowest ground state energy." It would be helpful if the author included more details on this - specifically, what configuration of bond operators D were used in this calculation? Is there a range of parameters where this exact two-fold degeneracy does not hold?

Re: I have added an appendix A to explain the details of the numerical confirmation. The configurations of bond operators D are chosen randomly within 2^N configurations that are related to the uniform configuration {D=1} by all possible local gauge transformations. For generic interaction strength U, the exact two-fold degeneracy still holds.

Action: An Appendix A “Numerical confirmation of the exact two-fold degeneracy” is added.

While written in terms of Majorana fermions, this model can also be written in terms of complex fermions coupled to a Z2 gauge field. Unless I misunderstand, I believe that this would give the BCS-Hubbard model studied in [47], coupled to a toric code. Regardless, I believe that expressing the model in this language at least once would be helpful to readers who are more familiar with such systems and also contextualize the model with related works. This also begs the question of why writing both the matter and gauge fields purely in terms of Majorana fermions is important. Solvability of the BCS-Hubbard model requires a Majorana description of the matter fields, but what utility does a Majorana description of the gauge fields give the reader?
The motivation for this representation should be made more clear in the paper - as keeping track of the four different flavors of Majorana fermions can be difficult to follow, it is important to have a clear motivation for why this notation is useful.
Re: This is a very crucial question about the Majorana fermion construction, which is related to locality, indistinguishability and symmetry.
1. The current interacting Majorana fermion model on square lattice is not a simple coupling between BCS-Hubbard model and toric code. Only in the infinite V limit, the interaction of \chi Majorna fermions is exactly identical to the toric code. However, the coupling between \gamma and \chi Majorna fermions will then become a nonlocal interaction between electrons of BCS-Hubbard model and spins of toric code, which violates the locality principle. Even though using the unfamiliar Majorana fermion representation, various limits are examined to make connection with related works
2. The realization of the model in conception is using the Majorana zero modes in vortex lattice. Thus it is natural and necessary to formulate the theory in terms of Majorana fermions. The Majorana fermion representation of matter fields, which puts spin and charge degrees of freedom on equal footing, provides a clear physical picture of intertwined orders. Writing gauge fields in terms of Majorana fermions provides a novel fractionalization routine of conventional bosonic gauge fields. Also the Majorana fermion representation put matter and gauge fields on equal footing. The Majorana fermion duality is explored in Ref. 40 and 59 according to the indistinguishability of Majorana fermions, it is possible to seek duality between matter and gauge fields in the Majorana fermion representation as in principle Majorana fermions are all identical particles.
3. For systems with 2n Majorana fermion flavors in total, the maximal symmetry of the system can be directly read out as SO(2n), such as Hubbard model at half-filling with SO(4) symmetry. Thus it is more straightforward to perform the symmetry analysis in terms of Majorana fermions.

Action: A paragraph is added in the Discussion section to highlight the motivation of Majorana fermion representation.

The author briefly mentions a topological phase transition at K/t=0, corresponding to the change between the Z2A and Z2B nature of the gauge fields, but does not elaborate on it further. I would suggest that more be said about this transition - is it clear what this transition is in the limit that the matter and gauge fields decouple? If so, is it unchanged by the presence of matter fields?
Re: The phase transition line K/t=0 separating Z2A and Z2B phases is of first order phase transition, akin to the magnetic field induced first order phase transition of Ising model across the zero magnetic field line. However, different from that the magnetic field breaks the Z_2 symmetry of Ising model, the plaquette interaction keeps the local Z_2 gauge symmetry of gauge fields. The first order phase transition is topological phase transition, which means the phase transition has nothing to do with symmetry but only the discrete gauge structure of topological order, i.e. Z_2 flux, changes across the phase transition line. The physics of symmetry breaking and intertwined orders of matter fields is unchanged across the phase transition line, and the only subtle changes are the quasiparticle dispersion and form factor due to the sign change of K. Such first order topological phase transition is unchanged in the presence of matter fields as matter and gauge fields still decouple in the infrared limit across the phase transition line.
Action: A paragraph is added at the end of V≠0 subsection to elaborate on the topological phase transition.

The author mentions the decoupling between matter and gauge degrees of freedom in the IR several times, but only shows this phenomenon explicitly in the limit U >> t. As I understand the paper, this feature is one of the central results, and a more explicit discussion of whether it hols for more general U/t would be helpful.
Re: For more generic parameters U and t, on the one hand, as the matter fields have discrete symmetry breaking, the low-energy excitations of matter fields are all gapped thus do not influence the gauge fields in the infrared limit, on the other hand, the matter fields only feel the gauge invariant quantities of gauge fields, i.e. the flux of plaquette, the plaquette interaction H_K fixes the Z_2 flux of gauge fields and the only low-energy excitations of gauge fields are gapped visons, thus do not influence the matter fields in the infrared limit. The decoupling of between matter and gauge degrees of freedom in the infrared limit is essentially unique to Majorana/Z_2 lattice gauge theory.

Action: A paragraph is added in the V≠0 subsection to discuss the decoupling of matter and gauge degrees of freedom in the infrared limit

In the large-U analysis on page 7, only positive U is considered. It would seem that a large and negative U limit is also accessible perturbatively - does this analysis agree with later results?
Re: The analysis of negative large U limit is the same as the positive large case. I have incorporated these two limits into one formula in Eq. 15.

Action: The Eq. 15 is generalized to include the negative large U limit.

In summary, the author presents a new and interesting model of interacting fermions which is believed to host a variety of non-trivial physics. A study of this model is a noteworthy contribution to the field and is suitable for publication. However, I found the presentation a little difficult to follow, and include some suggestions in "Requested changes" on how to improve readability.
Re: I have revised the paper in accordance with the reviewer’s suggestions. Hope this revised version will be more reader friendly.

Requested changes
1 - Typo at the end of page 13, "puryly" should be spelled "purely."
2 - On page 9, define more clearly what is meant by a "folded Brillouin zone."
3 - Include more details on the numerical diagonalization discussed in page 7.
4 - A clear motivation for why this Majorana representation of the gauge fields is important should be stated.
I would suggest the following changes in order to make the paper easier to follow; however, the author is free to disagree with this.

5 - The discussion of symmetries of the model, both local and global, should be presented at the very beginning of the paper when the Hamiltonian is introduced.

Re:
1- The typo is corrected.
2- The statement “…folded Brillouin zone” is revised to “…magnetic Brillouin zone, which is half of square lattice Brillouin zone.”.
3- An Appendix A is added to include more details of the numerical confirmation of exact two-fold degeneracy.
4- A paragraph is added in the Discussion to highlight the motivation of Majorana fermion representation.
5- The discussions of local and global symmetries of the model are moved to the very beginning of the paper when the Hamiltonian is introduced.

---

## Round 1 · Referee Report · Anonymous (Referee 2) · 2022-12-27

Strengths

The author presents and analyses a new and interesting Ising lattice gauge theory coupled to Majorana fermion matter. The Majorana representation utilised by the author seems to be a powerful tool for understanding the physics of this model.

Weaknesses

In the current form the paper is convoluted; no connections and comparisons to previous related literature is provided

Report

In the present manuscript the author constructs a basic Majorana Z_2 lattice gauge theory on a square lattice and uncovers salient features of its quantum phase diagram. I think the paper deserves to be published in SciPost Physics after the following suggestions are addressed by the author:

1) The presentation of the model can be greatly improved. For example, I strongly suggest the author to introduce Z_2 gauge redundancies (the Gauss law operator and how it acts on the building blocks of the model) directly after Eq. (8). This will define the model unambiguously. Currently this information is scattered throughout the paper. It would also be useful to introduce (already after Eq. (8)) global symmetries of the model that are relevant for later analysis.

2) Discrete gauge theories coupled to complex fermions have be studied actively recently, see e.g.

https://www.nature.com/articles/nphys4028

https://www.pnas.org/doi/abs/10.1073/pnas.1806338115

https://journals.aps.org/prx/abstract/10.1103/PhysRevX.10.041057

https://journals.aps.org/prb/abstract/10.1103/PhysRevB.105.075132

and references therein.

While the present paper considers a model without U(1) particle number conservation, it would be beneficial if the author compares and contrasts his main findings with the previous literature. In addition to the Majorana representation, it would be also appreciated if the model could be written in terms of complex fermions coupled to conventional Z_2 gauge fields that reside on lattice links. This will make the relation to previous studies more transparent.

3) One aspect of the U(1)-conserving models (mentioned above) on a square lattice which attracted much attention is the existence of the Z_2 deconfined phase at half filling with gapless Dirac fermions that originate from the pi-flux background of the Ising gauge fields. I suspect that a Majorana version of such gapless phase exists in the present model. Is it true? How can it be stabilised? How can gap it out?

4) Due to the Gauss law (14), the Majorana fermions $\gamma$ and $\chi$ are redundant gauge-variant degrees of freedom. It is possible to resolve the Gauss law (14) and express the model only in terms of the physical gauge-invariant Majorana fermions?

  • validity: good
  • significance: good
  • originality: high
  • clarity: ok
  • formatting: reasonable
  • grammar: good

Author:  Jian-Jian Miao  on 2023-01-14  [id 3236]

(in reply to Report 2 on 2022-12-27)

In the present manuscript the author constructs a basic Majorana Z_2 lattice gauge theory on a square lattice and uncovers salient features of its quantum phase diagram. I think the paper deserves to be published in SciPost Physics after the following suggestions are addressed by the author:

Re: I’m grateful for the referee’s constructive comments and suggestions. I have considered all the constructive suggestions made by the referee, enclosed please find my point-to-point response.

1) The presentation of the model can be greatly improved. For example, I strongly suggest the author to introduce Z_2 gauge redundancies (the Gauss law operator and how it acts on the building blocks of the model) directly after Eq. (8). This will define the model unambiguously. Currently this information is scattered throughout the paper. It would also be useful to introduce (already after Eq. (8)) global symmetries of the model that are relevant for later analysis.

Action: The discussions of local and global symmetries of the model are introduced directly after Eq. (8) when the full Hamiltonian is introduced.

2) Discrete gauge theories coupled to complex fermions have be studied actively recently, see e.g. https://www.nature.com/articles/nphys4028 https://www.pnas.org/doi/abs/10.1073/pnas.1806338115 https://journals.aps.org/prx/abstract/10.1103/PhysRevX.10.041057 https://journals.aps.org/prb/abstract/10.1103/PhysRevB.105.075132 and references therein. While the present paper considers a model without U(1) particle number conservation, it would be beneficial if the author compares and contrasts his main findings with the previous literature. In addition to the Majorana representation, it would be also appreciated if the model could be written in terms of complex fermions coupled to conventional Z_2 gauge fields that reside on lattice links. This will make the relation to previous studies more transparent.

Re: I have compared and contrasted with the previous literature on discrete gauge theories coupled to complex fermions and cited relevant references. In Eq. (2b), (5b), (12) and (13), the model has already been written in terms of complex fermions with equal-spin-pairing coupled to conventional Z_2 gauge fields that reside on lattice links.

Action: A paragraph is added in the Discussion section to compare and contrast with the previous literature and relevant references are cited therein.

3) One aspect of the U(1)-conserving models (mentioned above) on a square lattice which attracted much attention is the existence of the Z_2 deconfined phase at half filling with gapless Dirac fermions that originate from the pi-flux background of the Ising gauge fields. I suspect that a Majorana version of such gapless phase exists in the present model. Is it true? How can it be stabilised? How can gap it out?

Re: The quasiparticle dispersions of Majorana fermions are all gapped no matter in the 0-flux or pi-flux background of the Ising gauge fields. Thus the present model without U(1)-conserving does not support the exotic Z_2 deconfined phase with gapless Dirac fermions.

4) Due to the Gauss law (14), the Majorana fermions γ and χ are redundant gauge-variant degrees of freedom. It is possible to resolve the Gauss law (14) and express the model only in terms of the physical gauge-invariant Majorana fermions? Re: I have tried to resolve the Gauss law but so far can’t express the model only in terms of the physical gauge-invariant Majorana fermions. I have revised the paper in accordance with the reviewer’s other suggestions. Hope this revised version will be more reader friendly.

---

## Round 2 · List of Changes

(1) The manuscript is refitted into standard SciPost format.
(2) An Appendix A “Numerical confirmation of the exact two-fold degeneracy” is added.
(3) A paragraph is added in the Discussion section to highlight the motivation of Majorana fermion representation.
(4) A paragraph is added at the end of V≠0 subsection to elaborate on the topological phase transition.
(5) A paragraph is added in the V≠0 subsection to discuss the decoupling of matter and gauge degrees of freedom in the infrared limit.
(6) The Eq. 15 is generalized to include the negative large U limit.
(7) The statement of “…folded Brillouin zone” is revised to “…magnetic Brillouin zone, which is half of square lattice Brillouin zone ”
(8) The discussions of local and global symmetries of the model are moved to the very beginning of the paper when the Hamiltonian is introduced.
(9) A paragraph is added in the Discussion section to compare and contrast with the previous literature and relevant references are cited therein.
(2) An Appendix A “Numerical confirmation of the exact two-fold degeneracy” is added.
(3) A paragraph is added in the Discussion section to highlight the motivation of Majorana fermion representation.
(4) A paragraph is added at the end of V≠0 subsection to elaborate on the topological phase transition.
(5) A paragraph is added in the V≠0 subsection to discuss the decoupling of matter and gauge degrees of freedom in the infrared limit.
(6) The Eq. 15 is generalized to include the negative large U limit.
(7) The statement of “…folded Brillouin zone” is revised to “…magnetic Brillouin zone, which is half of square lattice Brillouin zone ”
(8) The discussions of local and global symmetries of the model are moved to the very beginning of the paper when the Hamiltonian is introduced.
(9) A paragraph is added in the Discussion section to compare and contrast with the previous literature and relevant references are cited therein.

---

## Editorial Decision

published